# Comparison of Surgical Resection and Percutaneous Ultrasonographic Guided Radiofrequency Ablation for Initial Recurrence of Hepatocellular Carcinoma in Early Stage following Curative Treatment

**DOI:** 10.3390/cancers14225524

**Published:** 2022-11-10

**Authors:** Hideko Ohama, Atsushi Hiraoka, Fujimasa Tada, Kanako Kato, Yoshiko Fukunishi, Emi Yanagihara, Masaya Kato, Hironobu Saneto, Hirofumi Izumoto, Hidetaro Ueki, Takeaki Yoshino, Shogo Kitahata, Tomoe Kawamura, Taira Kuroda, Yoshifumi Suga, Hideki Miyata, Masashi Hirooka, Masanori Abe, Bunzo Matsuura, Tomoyuki Ninomiya, Yoichi Hiasa

**Affiliations:** 1Gastroenterology Center, Ehime Prefectural Central Hospital, Matsuyama 790-0024, Japan; 2Department of Gastroenterology and Metabology, Ehime University Graduate School of Medicine, Toon 791-0295, Japan

**Keywords:** hepatocellular carcinoma, recurrence, early stage, surgical resection, radio frequency ablation

## Abstract

**Simple Summary:**

The SURF trial showed that surgical resection (SR) and radiofrequency ablation (RFA) had equal therapeutic effects for small hepatocellular carcinoma (HCC). This study aimed to elucidate therapeutic efficacy differences between SR and RFA for initial recurrent early-stage HCC. 371 patients with recurrent early-stage HCC after undergoing initial curative treatment were enrolled. Although the median albumin–bilirubin (ALBI) score was better in the SR than the RFA group (−2.90 vs. −2.50, *p* < 0.01), there were no significant differences between them in regard to RFS (median 28.1 months, 95% CI 23.4–50.0 vs. 22.1 months, 95% CI 19.3–26.2; *p* = 0.34), OS (78.9 months, 95% CI 49.3—not applicable vs. 71.2 months 95% CI, 61.8–84.7; *p* = 0.337), or complications (8.3% vs. 9.3%; *p* = 1.0). This retrospective study found equal therapeutic efficacy of SR and RFA for initial recurrence of early-stage HCC after undergoing curative treatment.

**Abstract:**

Background/Aim: The SURF trial showed that surgical resection (SR) and percutaneous ultrasonographic guided radiofrequency ablation (RFA) had equal therapeutic effects for small hepatocellular carcinoma (HCC). However, consensus regarding which treatment is appropriate for initial recurrent early-stage HCC remains lacking. This study aimed to elucidate therapeutic efficacy differences between SR and RFA for initial recurrent early-stage HCC. Materials/Methods: From 2000 to 2021, 371 patients with recurrent early-stage HCC (≤3 cm, ≤3 nodules) after undergoing initial curative treatment with SR or RFA were enrolled (median age 72 years; males 269; Child–Pugh A:B, *n* = 328:43; SR:RFA, *n* = 36:335). Recurrence-free survival (RFS) and overall survival (OS) were retrospectively evaluated. Results: Although the median albumin–bilirubin (ALBI) score was better in the SR than the RFA group (−2.90 vs. −2.50, *p* < 0.01), there were no significant differences between them in regard to RFS (median 28.1 months, 95% CI 23.4–50.0 vs. 22.1 months, 95% CI 19.3–26.2; *p* = 0.34), OS (78.9 months, 95% CI 49.3—not applicable vs. 71.2 months 95% CI, 61.8–84.7; *p* = 0.337), or complications (8.3% vs. 9.3%; *p* = 1.0). In sub-analysis for RFS and OS according to ALBI grade revealed no significant differences between the SR and RFA groups (ALBI 1/2 = 28.2/17.5 vs. 24.0/23.4 months; *p* = 0.881/0684 and ALBI 1/2 = 78.9/58.9 vs. 115.3/52.6 months, *p* = 0.651/0.578, respectively). Conclusion: This retrospective study found no significant differences in regard to RFS or OS between patients in the SR and the RFA groups for initial recurrence of early-stage HCC after undergoing curative treatment. These results showing equal therapeutic efficacy of SR and RFA confirm the findings of the SURF trial.

## 1. Introduction

In a report published in 2020, hepatocellular carcinoma (HCC) was shown to be the sixth most commonly diagnosed cancer worldwide and the third leading cause of cancer death [1]. Although surgical resection is usually the first choice treatment for early-stage HCC, the rate of recurrence within five years after that procedure ranges from 70–80% [2,3]. In addition, no strategy for the prevention of recurrence after curative treatment for HCC has been established; thus, it is clinically important to detect recurrence in an early stage and provide effective treatment to obtain a good prognosis.

Treatment strategies for HCC are based on liver function; tumor burden, including size and number; extrahepatic spreading; and vascular invasion [4]. According to the treatment algorithm of the Clinical Practice Guidelines for HCC presented by the Japan Society of Hepatology (4th edition), a hepatectomy procedure or radiofrequency ablation (RFA) is recommended for Child–Pugh class A/B patients with one to three tumors and a tumor diameter ≤3 cm without extrahepatic metastasis or vascular invasion [4]. The SURF trial, a randomized controlled trial conducted to compare outcomes of patients undergoing surgery with those undergoing RFA for small HCC, found no significant difference for median recurrence-free survival between a curative hepatectomy (3.0 years, 95% CI 2.6–5.1) and RFA (3.5 years, 95% CI 2.6–5.1) [5].

For patients with recurrent disease, intrahepatic recurrence has been reported to be the most common [6]. Although guidelines presented by European, American, and Eastern societies all recommend surgical resection as first-line treatment for naïve HCC [4,7,8], a recommendation for recurrent HCC cases has yet to be reported. Some studies have demonstrated a treatment algorithm for recurrent HCC, while only one randomized controlled trial has been reported, which found no statistically significant difference in survival outcome after a repeated surgical resection (SR) procedure as compared to RFA for patients with early-stage recurrent HCC [8]. However, supporting evidence and consensus regarding such a therapeutic algorithm for the treatment of recurrent HCC are lacking. The present study aimed to compare the therapeutic efficacy of SR with that of RFA as an initial treatment for recurrent early-stage HCC in patients who previously underwent curative treatment.

## 2. Materials and Methods

From January 2000 to December 2021, 371 patients with initial recurrent HCC in an early stage underwent SR or percutaneous ultrasonographic-guided RFA curatively as the initial treatment at Ehime Prefectural Central Hospital (median age 72 years, males 269, Child–Pugh A:B = 328:43) were enrolled, after exclusion of patients who did not undergo curative treatments as initial therapies and/or classified as Child–Pugh class C. Early-stage HCC was defined as the largest HCC diameter at 3 cm or less and the number of tumor nodules as three or less. The patients were divided according to treatment modality for recurrent HCC into the SR (*n* = 36) and RFA (*n* = 335) groups.

HCC diagnosis was primarily based on dynamic computed tomography (CT), Gd-EOB-DTPA-enhanced magnetic resonance imaging (EOB-MRI), and/or contrast-enhanced ultrasonography findings. Tumor diameter was determined using either early- or late-phase results. The tumor stage was defined according to the General Rules for the Clinical and Pathologic Study of Primary Liver Cancer [9]. Child–Pugh classification, [10] albumin–bilirubin (ALBI grade) [11,12], and modified ALBI (mALBI) grade, [13] for which ALBI grade 2 was divided into two sub-grades (mALBI 2a and 2b) using an ALBI score of −2.27 as the cut-off value, were used for hepatic function assessment.

SR was performed using either an open or laparoscopic approach. RFA was percutaneously performed under ultrasonography guidance to unstained areas appeared widely along the entire margin after RFA as compared to low-density areas seen in late-phase CT findings or as low-intensity areas in the hepatobiliary phase shown by MRI before RFA.

The endpoints were recurrence-free survival (RFS) and overall survival (OS). RFS was defined as the duration from the date of treatment for HCC to the date of diagnosis of recurrence, while OS was used as the duration from the date of starting treatment for recurrent HCC to the date of death for any reason. Adverse effects following treatments were recorded and examined.

## 3. Statistical Analysis

All data used were accessed from a database application. Statistically significant differences were analyzed using Student’s t-test, Welch’s t-test, and/or a Mann–Whitney U test, as appropriate. OS and RFS curves were generated using the Kaplan–Meyer method and compared with a log-rank test. The number of patients in the SR group was small; thus, even when the condition of conditional exchangeability was satisfied and the propensity score estimated correctly in a propensity score matching method, a bias may have been present. Therefore, inverse probability weighting (IPW) by propensity score was used to determine that the propensity scores for the present study were approximately the same using arbitrary criteria to prevent uncorrectable bias. Adjustment with IPW was performed in estimation for RFS and OS. Probabilities of the SR and RFA group were calculated by logistic regression analysis using covariates that may affect RFS and OS. In addition to clinical factors in the SR and RFA group with a *p*-value < 0.05, eastern cooperative oncology group performance status (ECOG PS) was used because all patients of the SR group were ECOG PS 0. For multivariate analysis, Cox proportional hazards model analysis was performed to assess hazard ratio (HR) and 95% confidence interval (CI). All statistical analyses were performed with EZR [14], ver. 1.53 (Saitama Medical Center, Jichi Medical University, Saitama, Japan), a graphical user interface for R (The R Foundation for Statistical Computing, Vienna, Austria), or more precisely, a modified version of R commander designed to add statistical functions frequently used in biostatistics.

## 4. Results

### 4.1. Patient Characteristics

The median age was 71 years [interquartile range (IQR): 65–74] in the SR group and 72 years (IQR: 64–78) in the RFA group. There were no significant differences regarding gender or etiology in either group. The median alanine aminotransferase level was higher in the RFA (32 U/L, IQR 21–54 U/L) than in the SR (21 U/L, IQR 16–38 U/L) group (*p* = 0.03). mALBI grade in the SR group was significantly better than that in the RFA group [−2.90 (IQR −3.04 to −2.58) vs. −2.50 (IQR −2.84 to −2.22; *p* <0.01]. There was also a significant difference regarding initial treatment between the groups, as 20 patients (55.5%) underwent SR in the SR group as compared to 112 (33.4%) in the RFA group (*p* = 0.01). Median tumor diameter was 1.6 cm (IQR 1.5–2.5) in the SR group and 1.5 cm (IQR 1.2–1.9) in the RFA group (*p* = 0.03), while there was no significant difference regarding tumor number between them (Table 1).

### 4.2. RFS and OS

The median post-enrollment follow-up period was 38.2 months (IQR 17.9–64.4 months) in the SR group and 46 months (IQR 23.1–73.7 months) in the RFA group, while the median RFS was not significantly different at 28.1 (95% CI 23.4–50.0) and 22.1 (95% CI 19.3–26.2) months, respectively, (*p* = 0.34) (Figure 1a). Furthermore, the median OS was 78.9 (95% CI 49.3—NA) and 71.2 (95% CI 61.8–84.7) months, respectively; also not significantly different (*p* = 0.337) (Figure 1b). When analysis after adjustment with IPW was performed, the median RFS was not significantly different between the SR group (23.4 months) and the RFA group (22.1 months) (*p* = 0.22). The median OS was 74.8 months and 72.2 months, respectively; not significantly different (*p* = 0.06). (Appendix A). For subgroup analysis, the patients were stratified by tumor size (≤2 vs. >2 cm), tumor number (solitary vs. multiple), age (<65 vs. ≥65 years), ALBI grade (1 vs. 2–3), and FIB4 index (≤2.6 vs. >2.6). RFS did not differ significantly between the SR and RFA groups after dividing into those subgroups (Figure 2a). OS was significantly better RFA with an FIB4 index ≤3.25, while SR with an FIB4 index >3.25 (*p* < 0.01) (Figure 2b).

### 4.3. Adverse Events

There was no significant difference in the rate of incidence of adverse events between the SR and RFA groups (8.3% vs. 9.3%, *p* = 1.0). In the SR group, adverse events developed in three patients, including bile leakage (*n* = 1, 2.8%), duodenal ulcer bleeding (*n* = 1, 2.8%), and abscess at the resection margin (*n* = 1, 2.8%), while adverse events were noted in 31 in the RFA group, with the most common including liver infarction (*n* = 6, 1.8%), pneumothorax (*n* = 5, 1.5%), dispersive electorode burn (*n* = 3, 0.9%), bleeding (*n* = 3, 0.9%), and acute cholecystitis (*n* = 2, 0.6%).

### 4.4. Sub-Analysis, RFS, and OS for each ALBI Grade

When OS and RFS were analyzed according to ALBI grade (Figure 3), the median RFS was 24 months (95% CI 19.1–28.5) for ALBI grade 1, 22.5 months (95% CI 19.3–27.6) for ALBI grade 2, and 44.6 months (95% CI 6.7—NA) for ALBI grade 3, which were not significantly different (*p* = 0.476). On the other hand, median OS was 115.3 months (95% CI 96.0—NA), 54.8 months (95% CI, 45.0–58.9), and 32.2 months (95% CI, 7.2–56.3), respectively, indicating significant differences among the three grades (*p* < 0.01). In comparisons of RFS and OS between the SR and RFA groups, the median recurrence-free period did not differ significantly between the SR and RFA patients classified as ALBI 1 (28.2 months, 95% CI 15.4–46.3 vs. 24.0 months, 95% CI 18.9–28.5; *p* = 0.881) (Figure 4a). After dividing based on ALBI grade, median OS was not significantly different between the SR and RFA patients classified as ALBI 1 (78.9 months, 95% CI 49.3—not applicable vs. 115.3 months, 95% CI 96.0—not applicable; *p* = 0.651) (Figure 4b). As for those classified as ALBI grade ≥ 2, the median RFS for the SR group was 17.5 months (95% CI 4.4–58.2) and for the RFA group was 23.4 months (95% CI 19.7–28.6); again, not a significant difference (*p* = 0.684) (Figure 5a). Median OS was also not significantly different, as that was 58.9 months (95% CI 7.8—not applicable) in the SR group and 52.6 months (95% CI 43.2–57.2) in the RFA group (*p* = 0.578) (Figure 5b).

## 5. Discussion

In the present patients, median RFS and OS in the SR group were 28.1 and 78.9 months, respectively, and 22.1 and 71.2 months, respectively, in the RFA group. A subgroup analysis showed that elevated FIB4 index was significantly related to OS, but when analysis after adjustment with IPW was performed, median RFS and OS were not significantly different between both groups. As treatment for the initial recurrence of early-stage HCC, patients who received RFA did not show a significant difference as compared to those who underwent an SR procedure for either RFS or OS.

SR has been shown to be a powerful curative treatment for HCC patients that contributes to a better prognosis, while recent improvements in laparoscopy technology have led to the indication of resection in greater numbers of cases [15]. Kaibori reported that median OS was no significant difference between the open (120 months, 95%CI 100—NA) and laparoscopic (75 months, 95%CI 52—NA) hepatectomy [16]. Although similar ninety-day mortality rates were seen in the two groups in this study, a recent meta-analysis showed that laparoscopic liver resection for recurrent HCC offered benefits in terms of lower in-hospital complication rates and blood loss levels, resulting in a shorter hospital stay as compared to open resection [17]. In the future, minimally invasive SR may also improve treatment outcomes. On the other hand, RFA has become established as the standard of care for patients with a small HCC tumor, including those considered unsuitable for SR. Newly developed ancillary techniques, e.g., virtual sonography (VUS) [18], contrast-enhanced sonography (CEUS) [19], and artificial pleural effusion and/or ascites, have contributed to increasing the feasibility of an RFA procedure and producing more reliable results. To achieve favorable local tumor control with percutaneous ablation therapy, accurate target tumor delineation by VUS and/or CEUS is considered to be very important [20]. Moreover, supplemental use of artificial pleural effusion and/or artificial ascites as part of the RFA procedure increases its safety and efficacy when treating tumors located near the diaphragm or surface of the liver [21].

Wang reported that repeat SR achieved the best outcomes for early intrahepatic recurrent HCC, while for patients with more invasive primary tumors and underlying cirrhosis with varices, repeat SR produced similar results as RFA [22]. Some retrospective non-randomized clinical studies compared long-term survival between repeat SR and RFA for early-stage recurrent HCC and reported that both showed equivalent outcomes [23,24,25,26,27]. Furthermore, Xia presented results of a randomized clinical trial in which no statistically significant differences in five-year OS or RFS were observed in comparisons between patients with early-stage recurrent HCC who underwent RFA and those who underwent repeat SR [28]. Although the ALBI score was worse in the present RFA group than the other, there were no significant differences in regard to RFS and OS between the RFA and SR therapeutic modalities; the same as noted in other studies. Another previous report suggested that RFA may be considered as first-line treatment for HCC up to 2 cm in diameter in patients with Child–Pugh class A or B liver cirrhosis [29]. Moreover, Zhou reported that RFA patients showed much shorter operative times and lower levels of blood loss and had shorter hospital stays than those that underwent SR [30]. Together, these findings indicate that RFA may provide a better prognosis for patients, even those with worse hepatic function; thus, this less invasive procedure might have a great role in patients whose hepatic function is slightly or obviously unfit for SR.

Starlinger reported that the combination of aspartate aminotransferase/platelet ration and ALBI was significantly associated with postoperative grade C liver dysfunction, thirty-day mortality, and liver dysfunction-associated thirty-day mortality and was superior to either score alone [31]. Tamaki reported that the rate of HCC development one, two, three, and four years after sustained virological response was significantly higher in patients with sustained FIB4 index >3.25 than in those whose FIB4 index decreased to ≤3.25 [32]. The severity of cirrhosis is considered a major factor in the recurrence of HCC. The genetic and epigenetic factors, such as microRNAs, were suggested to play a role in liver cirrhosis and its progression to HCC [33,34]. In this study, subgroup analysis showed that liver fibrosis was related to OS. The intensity of liver fibrosis may influence treatment decisions.

Previous studies have reported that ALBI grade, a tool for detailed assessment of hepatic function, is useful for predicting survival of HCC patients, with better baseline ALBI grade thought to be an important prognostic factor for OS [12,13]. In the present cohort, there was a significant difference in OS among ALBI grade classifications at the time of recurrence, regardless of the treatment modality used, whereas there were no significant differences in RFS or OS between the SR and RFA groups for each ALBI grade. We concluded that deciding which is the superior therapeutic modality is not as important as how to use available therapeutic equipment well in order to obtain a better prognosis for individual patients.

This study has some limitations, including its retrospective nature, as there was selection bias regarding treatment choice for recurrent HCC. In addition, the number of patients who underwent SR was too small, and the recent development of laparoscopic hepatectomy has made surgery less invasive, but this has not been taken into account because of its retrospective nature. In the near future, a randomized control trial with a larger number of patients who underwent SR will be needed to obtain more accurate conclusions for the proper use of SR and RFA for recurrent early-stage HCC.

## 6. Conclusions

The present findings suggested that RFS and OS were not different between the SR and RFA groups after classification as ALBI 1 or 2/3. They are considered to demonstrate that these two modalities have equivalent effects on RFS and OS when used to treat patients with initial recurrent HCC at an early stage, i.e., tumor size ≤3 cm and ≤3 nodules.

## Figures and Tables

**Figure 1 cancers-14-05524-f001:**
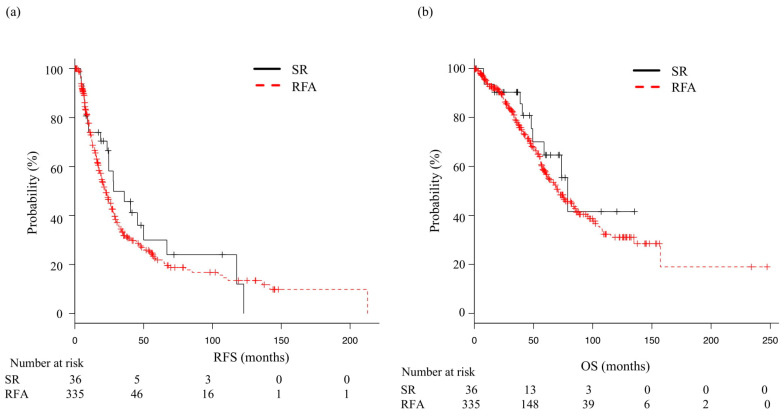
Recurrence-free and overall survival in patients who underwent surgical resection (SR) or radiofrequency ablation (RFA). (**a**) Median recurrence-free survival (mRFS) in the SR and RFA groups was 28.1 (95% CI: 23.4–50.0) and 22.1 (95% CI: 19.3–26.2) months, respectively (*p* = 0.34). (**b**) Median overall survival (mOS) in the SR and RFA groups was 78.9 (95% CI: 49.3—not applicable) and 71.2 (95% CI: 61.8–84.7) months, respectively (*p* = 0.337).

**Figure 2 cancers-14-05524-f002:**
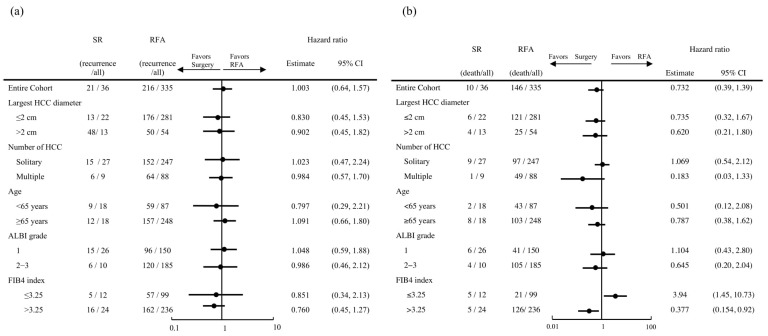
Subgroup analysis of recurrence-free survival (RFS) and overall survival (OS) in surgical resection (SR) and radiofrequency ablation (RFA) groups. (**a**) RFS did not differ significantly between the SR and RFA groups after dividing into subgroups. (**b**) OS was significantly better RFA with FIB4 index ≤ 3.25, while SR with FIB4 index > 3.25 (*p* < 0.01).

**Figure 3 cancers-14-05524-f003:**
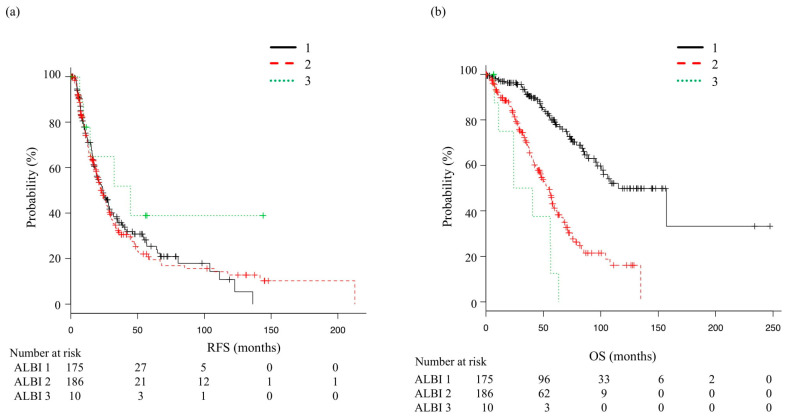
Recurrence-free survival (RFS) and overall survival (OS) according to albumin–bilirubin (ALBI) grade. (**a**) Median recurrence-free survival (mRFS) was 24 months (95% CI: 19.1–28.5) for ALBI grade 1, 22.5 months (95% CI: 19.3–27.6) for ALBI grade 2, and 44.6 months (95% CI 6.7—not applicable) for ALBI grade 3 (*p* = 0.476). (**b**) Median OS was 115.3 months (95% CI: 96.0—not applicable) for ALBI grade 1, 54.8 months (95% CI: 45.0–58.9) for ALBI grade 2, and 32.2 months (95% CI, 7.2–56.3) for ALBI grade 3 (*p* < 0.01).

**Figure 4 cancers-14-05524-f004:**
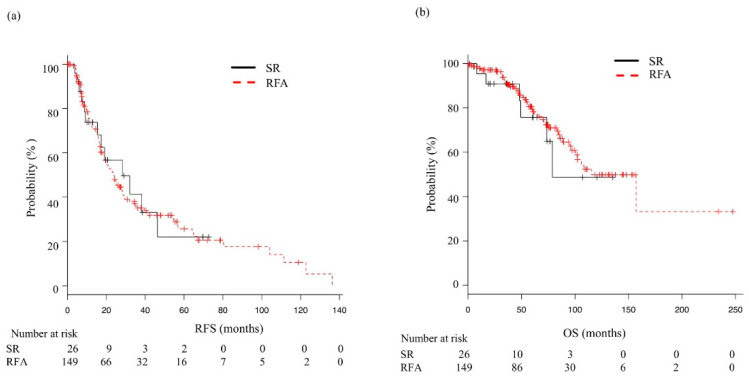
Recurrence-free survival (RFS) and overall survival (OS) following surgical resection (SR) or radiofrequency ablation (RFA) in albumin–bilirubin (ALBI) grade 1 patients. (**a**) Median RFS in the SR and RFA groups was 28.2 (95% CI: 15.4–46.3) and 24.0 (95% CI: 18.9–28.5) months, respectively (*p* = 0.881). (**b**) Median OS in the SR and RFA groups was 78.9 (95% CI: 49.3—not applicable) and 115.3 (95% CI: 96.0—not applicable) months, respectively (*p* = 0.651).

**Figure 5 cancers-14-05524-f005:**
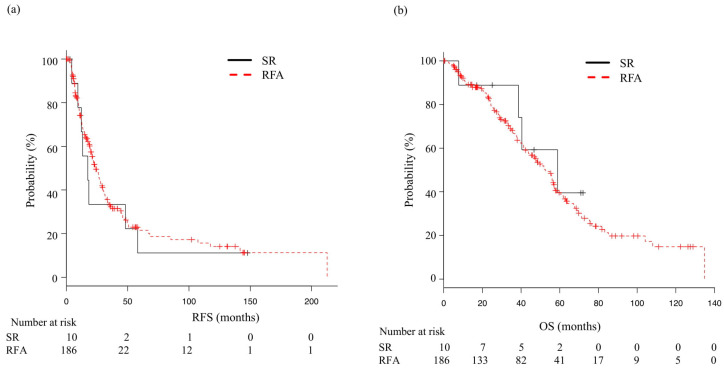
Recurrence-free survival (RFS) and overall survival (OS) following surgical resection (SR) or radiofrequency ablation (RFA) in patients with albumin–bilirubin (ALBI) grade ≥ 2. (**a**) Median RFS in the SR and RFA groups was 17.5 (95% CI: 4.4–58.2) and 23.4 (95% CI: 19.7–28.6) months, respectively (*p* = 0.684). (**b**) Median OS in the SR and RFA groups was 58.9 (95% CI: 7.8—not applicable) and 52.6 (95% CI: 43.2–57.2) months, respectively (*p* = 0.578).

**Table 1 cancers-14-05524-t001:** Characteristics of patients (*n* = 371).

	SR Group (*n* = 36)	RFA Group (*n* = 335)	*p*-Value
Age, years *	71 (65–74)	72 (64–78)	0.29
Gender, males:females	26:10	243:92	0.97
Etiology, HBV:HCV (SVR):alcohol:others	4:25 (8):3:4	32:226 (32):24:53	0.87
ECOG PS, 0:1:2:3	36:0:0:0	322:11:1:1	0.321
AST, U/L *	28 (23–40)	42 (29–62)	<0.01
ALT, U/L *	21 (16–38)	32 (21–54)	0.03
Platelets, 10^4^/µL *	12.3 (9.2–15.1)	11.3 (7.9–15.6)	0.39
Total-bilirubin, mg/dL *	0.8 (0.6–1.0)	0.7 (0.5–1.0)	0.87
Albumin, g/dL *	4.3 (3.9–4.4)	3.8 (3.5–4.2)	<0.01
Prothrombin time, % *	90.5 (77.9–101)	85 (75.3–94)	<0.01
mALBI 1:2a:2b:3	25:5:3:1	147:9 2: 84:9	<0.01
FIB4 index *	3.78 (2.60–4.60)	4.62 (2.97–7.74)	0.01
Child–Pugh score A:B	33:3	295:40	0.46
AFP, ng/mL *	9.8 (4.0–103.5)	9.3 (4.2–29.9)	0.43
DCP, mAU/mL *	61 (25–146.5)	28.5 (19–62.5)	<0.01
Initial curative treatment (SR:RFA)	20:16	112:223	0.01
Tumor size (maximum), cm *	1.6 (1.5–2.5)	1.5 (1.2–1.9)	0.03
Tumor number (single:multiple)	27:9	247:88	1.0

* Median. Values in parentheses show interquartile range unless otherwise indicated. SR: surgical resection, RFA: radiofrequency ablation, HBV: hepatitis B virus, HCV: hepatitis C virus, SVR: sustained virological response by anti-viral treatments, ECOG PS: Eastern cooperative oncology group performance status, AST: aspartate aminotransferase, ALT: alanine aminotransferase, mALBI grade: modified albumin–bilirubin grade, AFP: alpha-fetoprotein, DCP: des-gamma-carboxy prothrombin.

## Data Availability

Due to the nature of this research, the participants could not be contacted regarding whether the findings could be shared publicly; thus, supporting data, including datasets generated and/or analyzed for the current study, are not publicly available.

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
