# Peer review of "Comparison of Surgical Resection and Percutaneous Ultrasonographic Guided Radiofrequency Ablation for Initial Recurrence of Hepatocellular Carcinoma in Early Stage following Curative Treatment"

_cancers, 2022, doi:10.3390/cancers14225524_

Round 1
Reviewer 1 Report
Ohama and colleagues present results from a retrospective study on outcome of comparing surgical resection and RFA for recurrent HCC. They show that both arms are comparable in terms of RFS and OS regardless of mALBI score, tumor size and tumor numbers. The study is well conducted, and results are well presented and discussed. There are only a few minor comments about this study.
1. The severity of cirrhosis is considered a major factor in late recurrence in early-stage HCC after curative treatment. Also, recent study showed that APRI was shown to be effective in predicting prognosis after surgical resection (https://doi.org/10.1093/bjsopen/zraa043). Since AST level differ between SR and RFA group, and Child A consists of cirrhotic changes of varying severity, it would be even more interesting to compare APRI between the two groups using subgroup analysis.
2. In line 159, it needs to be revised to ‘44.6 months (95% CI 6.7-NA) for ALBI grade 3’
3. In line 162-64, authors should add which subgroup it was performed. Probably in mALBI grade 1.
4. ECOG is an important factor when choosing between SR and RFA for curative treatment. It would be much better if the authors present ECOG level in Table 1 and if possible, to perform a subgroup analysis regarding ECOG level.
Author Response
Response to Reviewer 1 Comments
- The severity of cirrhosis is considered a major factor in late recurrence in early-stage HCC after curative treatment. Also, recent study showed that APRI was shown to be effective in predicting prognosis after surgical resection (https://doi.org/10.1093/bjsopen/zraa043). Since AST level differ between SR and RFA group, and Child A consists of cirrhotic changes of varying severity, it would be even more interesting to compare APRI between the two groups using subgroup analysis.
Response1.As your indicated, the severity of cirrhosis is a major factor in late recurrence in early-stage HCC after curative treatment. However, FIB4 index is more commonly used than APRI as an index of fibrosis in Japan. Therefore, subgroup analysis was performed using the FIB4 index.
- In line 159, it needs to be revised to ‘44.6 months (95% CI 6.7-NA) for ALBI grade 3’
Response 2. Thank you for pointing this out. I have corrected it as you indicated.
- In line 162-64, authors should add which subgroup it was performed. Probably in mALBI grade 1.
Response 3. Thank you for pointing this out. It was performed in ALBI grade1. I have corrected it as above.
- ECOG is an important factor when choosing between SR and RFA for curative treatment. It would be much better if the authors present ECOG level in Table 1 and if possible, to perform a subgroup analysis regarding ECOG level.
Response 4. Thank you for pointing this out. I have also included ECOG-PS in Table1. Subgroup analysis was not performed because all patients in the SR group was PS0.
Reviewer 2 Report
GENERAL COMMENT: This is an interesting retrospective clinical study to assess the results of two very different techniques to treat recurrence of liver cancer: surgical resection vs. RF ablation. Overall, the article is well written, the results well presented, and the methodology well described. My greatest concern is related to the conclusions and the level of clinical evidence that can be derived from the results, all due to the limitations recognized by the authors themselves. The authors must state in abstract and conclusion the level of evidence about the conclusions.
Other issues:
1) The number of patients in each group is very different, so the reason should be justified and explain how this discrepancy impacts the statistical analysis.
2) Consider explaining a little more about the characteristics and the possible cause of the burn in 3 patients in the RFA group.
3) Did the author think using other statistical techniques such as propensity score matching to compare the results between groups? See e.g. Song KD, Lim HK, Rhim H, et al. Repeated Hepatic Resection versus Radiofrequency Ablation for Recurrent Hepatocellular 318 Carcinoma after Hepatic Resection: A Propensity Score Matching Study. Radiology. 2015; 275: 599-608. Explain why these techniques were not used.
4) In order to be able to draw conclusions that suggest a longer-term clinical strategy, I suggest that the authors attempt to conduct a further analysis from four groups depending on the strategy used to treat the recurrence (RFA vs. resection) and the used approach at the first treatment (RFA vs. resection), i.e. a total of four groups, obviously, if the resultant sample sizes are large enough.
5) How the authors think the retrospective nature of the study might impact the results? Indicate it in the limitation section. For example, throughout these 10 years, were there significant changes in the procedural and technical aspects of both groups?
6) As the authors themselves acknowledge, the limited number of patients in the surgical resection group is a major limitation, so I suggest that the conclusions be rewritten in more modest terms, e.g. The present findings suggest, instead of show...
Author Response
Response to Reviewer 2 Comments
1) The number of patients in each group is very different, so the reason should be justified and explain how this discrepancy impacts the statistical analysis.
Response 1. Thank you for pointing this out. I performed adjustment with inverse probability weighting was used to prevent any uncorrectable bias by setting an arbitrary criterion to determine that the propensity scores in the present study were approximately the same.
2) Consider explaining a little more about the characteristics and the possible cause of the burn in 3 patients in the RFA group.
Response 2. Their burns occurred at the site of the counterplate. I had added that to the manuscript.
3) Did the author think using other statistical techniques such as propensity score matching to compare the results between groups? See e.g. Song KD, Lim HK, Rhim H, et al. Repeated Hepatic Resection versus Radiofrequency Ablation for Recurrent Hepatocellular 318 Carcinoma after Hepatic Resection: A Propensity Score Matching Study. Radiology. 2015; 275: 599-608. Explain why these techniques were not used.
Response 3. Thank you for pointing this out. I have additionally considered with adjustment with inverse probability weighting.
4) In order to be able to draw conclusions that suggest a longer-term clinical strategy, I suggest that the authors attempt to conduct a further analysis from four groups depending on the strategy used to treat the recurrence (RFA vs. resection) and the used approach at the first treatment (RFA vs. resection), i.e. a total of four groups, obviously, if the resultant sample sizes are large enough.
Response 4. Thank you for pointing this out. If the patients separate to four groups, the sample size is too small. Therefore, initial treatment was used to adjustment with inverse probability weighting.
5) How the authors think the retrospective nature of the study might impact the results? Indicate it in the limitation section. For example, throughout these 10 years, were there significant changes in the procedural and technical aspects of both groups?
Response 5. Recently, laparoscopic hepatectomy has become possible, and surgery is expected to become less invasive in the future. We have not been able to examine this issue because of the retrospective study and have added it to the limitation section.
6) As the authors themselves acknowledge, the limited number of patients in the surgical resection group is a major limitation, so I suggest that the conclusions be rewritten in more modest terms, e.g. The present findings suggest, instead of show...
Response 6. Thank you for pointing this out. I have corrected it as you indicated.

Round 2
Reviewer 2 Report
Just to change "counterplate" to "dispesive electrode" (or patch)